# Sound Stimulation Can Affect *Saccharomyces cerevisiae* Growth and Production of Volatile Metabolites in Liquid Medium

**DOI:** 10.3390/metabo11090605

**Published:** 2021-09-07

**Authors:** Alastair Harris, Melodie A. Lindsay, Austen R. D. Ganley, Andrew Jeffs, Silas G. Villas-Boas

**Affiliations:** 1School of Biological Sciences, University of Auckland, 3A Symonds Street, Auckland 1010, New Zealand; ahar394@aucklanduni.ac.nz (A.H.); melodie.lindsay@auckland.ac.nz (M.A.L.); a.ganley@auckland.ac.nz (A.R.D.G.); a.jeffs@auckland.ac.nz (A.J.); 2Luxembourg Institute of Science and Technology, 5 rue Bommel, Z.A.E. Robert Steichen, L-4940 Luxembourg, Luxembourg

**Keywords:** yeast, sonic vibration, sonic pressure, aromas, Solid-Phase Microextraction (SPME), beverage, wine, beer, metabolism

## Abstract

The biological effect of sound on microorganisms has been a field of interest for many years, with studies mostly focusing on ultrasonic and infrasonic vibrations. In the audible range (20 Hz to 20 kHz), sound has been shown to both increase colony formation and disrupt microbial growth, depending upon the organism and frequency of sound used. In the brewer’s yeast *Saccharomyces cerevisiae*, sound has been shown to significantly alter growth, increase alcohol production, and affect the metabolite profile. In this study, *S. cerevisiae* was exposed to a continuous 90 dB @ 20 μPa tone at different frequencies (0.1 kHz, 10 kHz, and silence). Fermentation characteristics were monitored over a 50-h fermentation in liquid malt extract, with a focus on growth rate and biomass yield. The profile of volatile metabolites at the subsequent stationary phase of the ferment was characterised by headspace gas chromatography–mass spectrometry. Sound treatments resulted in a 23% increase in growth rate compared to that of silence. Subsequent analysis showed significant differences in the volatilomes between all experimental conditions. Specifically, aroma compounds associated with citrus notes were upregulated with the application of sound. Furthermore, there was a pronounced difference in the metabolites produced in high- versus low-frequency sounds. This suggests industrial processes, such as beer brewing, could be modulated by the application of audible sound at specific frequencies during growth.

## 1. Introduction

Sound is a mechanical vibration that propagates through elastic media, such as liquid, air, and solids. Sound can comprise varying frequencies, with the audible frequency range in air for humans commonly accepted as between 20 hertz (Hz) and 20 kHz (although this range is significantly reduced with age) [1]. Sound, across the entire spectrum, has been shown to influence organisms with traditional auditory pathways, such as humans, and organisms without auditory pathways, such as a variety of plants [2,3]. Vibrations below and above the audible range, infrasound and ultrasound, respectively, have been shown to significantly affect the growth and development of microorganisms [4,5,6,7,8]. However, little is known about the effect of audible sound range on microbial cells [9].

Studies that have investigated audible sound effects on microbes have typically found impacts on the growth and metabolism of model microorganisms. For example, three sound stimuli (1, 5, and 15 kHz pure tones) increased the biomass yield of *Escherichia coli* compared to a silent control, with a 5 kHz pure tone producing the largest change [10]. Similarly, another study found that three different sound stimuli (1, 5 and 10 kHz pure tones) promoted *E. coli* growth under normal and stressed conditions, with a significant reduction in the inhibitory effect of 3% sodium chloride [11]. In our previous work with the yeast *Saccharomyces cerevisiae*, two continuous pure tones (0.1 and 10 kHz) at an intensity of ~90 dB increased growth rate during exponential phase by an average of 12% compared to silence and altered the levels of 32 metabolites [12]. Furthermore, classical Indian music was shown to increase *S. cerevisiae* growth by ~3% compared to silence [13].

It has even been reported that the bacterium *Bacillus carboniphilus* emits very low-intensity sound that affects the growth of neighbouring colonies, significantly increasing their colony-forming efficiency under heavy metal stress conditions [14]. Sound applied to these microorganisms by loudspeakers mimicked the growth-promoting effect of this microbial sound [15]. Similarly, *S. cerevisiae* has been shown to produce nanomechanical motion in the range of 0.9–1.6 kHz, depending on the temperature of growth [16]. These vibrations are sufficiently strong (~10 nN) to affect the cell membrane, propagating into the surrounding media, and can be mimicked by artificial sound waves.

To understand how sound affects microbes requires an understanding of the pathways, cellular changes, and other underlying mechanisms driving a microbial response. However, due to the complexity of sound, these mechanisms could be extremely diverse, and as such a well characterised model organism is required [17]. *S. cerevisiae* is the preferred eukaryotic single cell model organism, with a well characterised protein network and metabolic response in many conditions [18,19]. Furthermore, *S. cerevisiae* is extensively used in industrial processes [20]. As such, *S. cerevisiae* is the perfect organism to extend our understanding of the effect of sound on microorganisms, from the previous studies conducted in *E. coli*, to an industrially relevant eukaryotic organism.

The metabolism of microorganisms is important for many industrial processes, including the production of a wide variety of food products and high-value molecules [21]. Of these processes, the production of alcoholic beverages through fermentation by *S. cerevisiae* remains one of the largest industrial uses of microorganisms [20,22]. In the beer industry, the fermentation of malt to ethanol is the major role of *S. cerevisiae*. However, the production of secondary metabolites by yeast, such as higher alcohols, esters, and phenols, impacts the flavour profile of beer [23]. In particular, volatile compounds provide a major proportion of the sensory profile of beer. The compounds produced are determined by the strain of yeast used, growth rate, and environmental conditions during fermentation [24,25]. Many of these volatile compounds are produced during the exponential growth phase of fermentation [26,27]. Consequently, applying sound to alter the growth rate, thereby altering the length of the exponential growth phase, may result in changes to the volatile metabolite composition produced during fermentation.

In this study, we tested the effect of audible sound on *S*. *cerevisiae* growth and volatile metabolite production during liquid fermentation. Both the medium, produced in bulk to limit the effect of environmental differences, and the strain were selected to imitate the production of beer. We found that sound increased the growth rate and altered the culture headspace volatilome during fermentation. These results suggest that application of audible sound during beverage fermentation has the potential to modify the resulting flavour.

## 2. Results

### 2.1. Audible Sound Affects the Growth of S. cerevisiae during Liquid Fermentation

The most commonly reported effect of audible sound on microorganisms is increased growth rate compared to cultures in silence [10,11,12,13,28,29,30,31]. Therefore, we measured the growth rate of *S. cerevisiae* strain CLIB382 on malt extract broth exposed to two sound stimuli (low-frequency at 100 Hz and high-frequency at 10 kHz) and compared this growth profile to fermentations without sound (silence). All fermentations were conducted in series in our Sound Isolation Chamber (See Section 4).

Biomass was measured using OD_600_ at several time points throughout the fermentations and was compared between sound conditions. Following the end of the lag phase (16 h), there was significantly greater biomass in fermentations exposed to both sound treatments compared to those treated with silence (Figure 1; *p* < 0.001). Furthermore, there was a significantly higher OD_600_ observed in fermentations exposed to low-frequency sound (0.893 ± 0.07, 100 Hz) compared to high-frequency (0.683 ± 0.02, 10 kHz, *p* < 0.01, Figure 1). However, the biomass at the end of the fermentations (33 h) was not significantly different between cultures exposed to both sound treatments and silence (*p* > 0.05). We compared the maximum growth rates, calculated from maximum change in biomass, in the fermentations during the exponential growth phase. The maximum growth rate of cultures grown without sound stimulus was 0.137 h^−1^ ± 0.005, which was lower than the maximum growth rates of the cultures exposed to low- and high-frequency sound (0.155 h^−1^ ± 0.008 and 0.177 h^−1^ ± 0.011, respectively). However, only the maximum growth rate of cultures exposed to high-frequency sound showed statistical significance when compared to the control cultures exposed to silence (*p* < 0.05). Nonetheless, the length of the exponential growth phase was reduced to 26.5 h under both sound treatments compared to 34.0 h in the silence control.

### 2.2. Profile of Volatile Metabolites

The profile of volatile metabolites in the headspace of the fermentation broths is of particular importance to the food industry, imparting much of the flavour perception to fermented beverages and foods. Many of these compounds are produced during the exponential microbial growth phase [26]. Therefore, we investigated whether the effects on yeast growth from applying audible sound would impact the profile of volatile compounds produced during fermentation. To test this, the volatile compounds produced under the two sound treatments were compared with the silence control. We putatively identified a total of 522 compounds in the headspace of the samples analysed using the NIST 2017 mass spectral library. Among the 522 volatile compounds, 346 were present at significantly different levels among the three different sound treatments (Appendix A; *p* < 0.05).

Analysing the fold change differences between treatments revealed that the exposure to sound appears to have significantly affected the development of aroma-related compounds during fermentation. A total of 24 of the significantly altered volatile metabolites were identified as common aroma compounds found in fermented beverages (Appendix B; [32]). Of these, the levels of citrus-related aromas, in particular, showed overall increases (Figure 2). For instance, limonene (the orange/citrus peel aroma) was detected at levels 7.8-fold higher in the high-frequency sound treatment compared to the silence control. However, compounds associated with sweet fruits, such as apricot aroma (ethyl octanoate), were significantly reduced under both high- and low-frequency sound treatments compared to silence (Figure 2). While most of the significant differences occurred between the two sound treatments and silence, some differences were also observed between the high- and low-frequency sound treatments. For example, production of benzoin by *S. cerevisiae* was significantly (*p* < 0.001) elevated in ferments exposed to high-frequency compared to low-frequency sound treatment (Figure 2).

To see if we could distinguish samples from different sound treatments based on the profile of aroma-associated compounds, we projected the levels of these aroma compounds from the three treatments onto a lower-dimensional space using linear discriminant analysis (LDA; Figure 3). LDA revealed very clear separations between each sound treatment, with 100% accuracy based on leave-one-out cross-validation (Table 1). The most significant compounds driving this separation (R-squared ≥ 0.9) were the aroma building block *cis*-2-nonene, and the aroma compounds 1-nonanol and *p*-cymene, associated with citronella and cumin/thyme, respectively (Appendix C).

## 3. Discussion

In this study, we demonstrated that sound impacts the fermentation performance of brewer’s yeast *S. cerevisiae* CLIB382. Specifically, the growth rate was increased, particularly through the application of high-frequency sound, and the levels of many volatiles were altered in the headspace of the fermentation. Interestingly, the production of several volatile compounds associated with flavour and aroma was affected differently, depending on the frequency of sound applied to fermentations. These results are relevant to industrial fermentation applications that require both high throughput batch cultures and specific flavour profiles, such as the brewing industry.

Our results are consistent with several studies that have shown that exposure to sound, compared to a silent control, results in increases in the rate of growth and/or biomass production of plants, bacteria, and yeasts [2,3,33,34]. In microbes, the magnitude of change in observed growth varies significantly, with the growth rate increase varying from 3.2% in *S. cerevisiae* to more than 300% in *Brevibacterium* sp. [13,33]. However, it is unclear how the growth is affected or why the response varies so dramatically between species. In contrast, a previous study employing *S. cerevisiae* [12] showed decreased biomass production following exposure to sound, whereas our study showed no difference in the final biomass concentration between fermentations exposed to either high- or low-frequency sound or silence. This previous study found an increased growth rate of the yeast in response to sound exposure, similar to ours, albeit with a lower magnitude (~3–10% compared to 28.9% for the high-frequency sound treatment in our study). The previous study was performed with minimal medium, versus the rich medium we employed, and the yeast strain (beer versus wine), aeration (anaerobic versus aerobic), agitation, and vessel (stirring flasks versus stationary tubes) were different, all of which could affect overall biomass production.

The composition of volatiles produced during fermentation was shown to be markedly altered by the application of sound in our study. The production of these compounds is directly impacted by both yeast metabolism and the rate of fermentation. For example, increased yeast growth has been shown to increase the production of fusel oils and sulphite compounds [34]. Therefore, it is not unexpected that an increased growth rate due to sound could result in changes to the volatile fraction. Furthermore, physiological stress can also alter the production of many of these compounds [35]. Sound, which produces changing zones of high and low pressure, may directly stress the cellular membrane, causing cells to flex and thereby potentially increasing cell membrane tension. Physiological stress on the cell membranes may, in turn, alter the production of secondary metabolites, such as volatiles. However, the mechanism(s) through which sound produces the effects we report has yet to be determined. In *E. coli*, sound was shown to increase the motility, particularly by treatment with high-frequency sound, which could increase access to nutrients and hence increase growth [36]. Furthermore, sound was shown to affect the intracellular RNA production [37], which could drive some changes in metabolism, but changes in the yeast transcriptome have yet to be determined.

Volatile compounds directly impact the flavour profile of fermentation beverages, such as beer. Recently, it was shown that sound can significantly impact various sensory descriptors, such as foam height, in beer fermentations [38]. This study used low frequencies, varying from 20–75 Hz, cycling across the frequency range each minute during the experiment. However, a sensory panel (n = 10) was unable to identify differences in the flavour profile of the beers produced compared to silence. This may suggest that the differences we observed in the range of volatiles produced do not alter the flavour of a beverage sufficiently to be perceived by the human olfactory–gustatory systems. However, the sound in this previous study was applied during the maturation of the beer (secondary fermentation), rather than during primary fermentation, which makes a direct comparison of the results with our study less certain.

The volatile compounds present in beer are derived from hops, malt, microorganisms (predominantly yeast), and adjuncts [39]. In our study, the medium and yeast strain were designed to mimic brewing; however, hops were not included to reduce the complexity of the medium. Even with this limitation, we demonstrated that sound can affect the production of many citrus- and floral-associated aroma compounds, such as limonene, geraniol, and linalool. These compounds have previously been associated with hop-derived compounds and biotransformation carried out by yeast during fermentation [40]. As such, it is likely that the application of sound has significant effects on other hop-derived compounds. Furthermore, our study demonstrated the frequency of sound can also impact this process, with the production of geraniol ~1.8-fold higher in fermentations exposed to high-frequency sound compared to low-frequency.

In wine, the effect of non-*Saccharomyces* strains on the production of volatile compounds has recently received greater attention [41]. These organisms have been shown to positively influence sensory quality through their unique metabolic processes. Presumably, sound could similarly influence the metabolism of these organisms, resulting in even greater influence on the sensory profile than suggested by this study. Many of these strains enter the wine fermentation from the vineyard rather than via purposeful addition by the winemaker [42]. Recently, a study showed that sound could alter the native plant microbiome in the vineyard [43]. This could further drive changes in the fermentation profile by altering the starting composition of ferments. The difference in volatile production could be driven by changes in the metabolism of yeast during exponential growth. For instance, it has been demonstrated that low-frequency sounds can increase the activity in numerous yeast metabolic pathways, whereas high-frequency sound depressed the activity of pathways associated with aromatic amino acid biosynthesis [12]. However, this previous study analysed the change in intracellular metabolites during exponential growth, whereas our volatile metabolites were measured in samples harvested the during post/late-exponential growth phase, which is a more metabolically heterogeneous phase of growth [44,45]. Nonetheless, together, the results of these studies suggest sound could be used to modulate *S. cerevisiae* metabolism and growth, albeit dependent on the general fermentation and growth conditions. Methods of online monitoring of volatile organic compounds have the potential to be effectively deployed to determine changes in the metabolism of yeast during the various growth phases [25,46].

## 4. Materials and Methods

### 4.1. Yeast Strain

*Saccharomyces cerevisiae* strain CLIB382, isolated from an Irish brewery in 1950, was maintained at 28 °C on malt extract agar (MEA) plates containing malt extract (20 g·L^−1^); dextrose (20 g·L^−1^); peptone (6 g·L^−1^); and agar (15 g·L^−1^) at pH 5.5.

### 4.2. Malt Extract Micro-Fermentations

The strain of *S. cerevisiae* was grown anaerobically in capped 13-mL polypropylene test tubes (Sarstedt Inc, Nümbrecht, Germany) containing 10 mL malt extract broth (MEB, as described above without agar). A bulk volume of MEB was prepared to avoid composition differences between tubes. MEB (250 mL) was inoculated with one colony of *S. cerevisiae* and grown at 28 °C under constant agitation (160 rpm, 48 h). This inoculum was pelleted, the medium discarded, and the pellet used to inoculate a fresh aliquot of MEB (250 mL) to an optical density of 0.1 at 600 nm (OD_600_) (measured with a Hitachi (model U-1100) spectrophotometer). Then, 10 mL aliquots were aseptically transferred into the test tubes. Test tubes were placed in an experimental soundproof chamber (Figure 4) inside a temperature-controlled room (28 °C). Sound was applied into the chamber by a Logitech LS-21 loudspeaker set connected to a laptop using VLC media player (version 2.2.8) to play the sound file. The liquid cultures were incubated in the chamber without agitation and under constant sound treatment until reaching stationary phase.

#### 4.2.1. Sound Treatments

Two sound treatments and two silence controls were performed. For each treatment, 21 replicate fermentations were prepared. The sound treatments were a continuous pure tone of low-frequency (100 Hz), and one of high-frequency (10 kHz), both at the same intensity (90 dB @ 20 μPa), measured at the location of the fermentation tubes using a Digitech QM1592 professional sound meter. The background noise level in the sound chamber was 41 dB @ 20 μPa when the speakers were not activated, and the results of the two silence controls were combined for further analysis.

#### 4.2.2. Fermentation

Sampling began 16 h after inoculation and continued every 4 h until the stationary growth phase was reached (after ~40 h). At each time point, three microfermentation tubes were harvested and OD_600_ measured. Contamination was assessed using light microscopy.

The specific growth rate (µ) was calculated as:(1)$$µ=\frac{2.303\left(\log\left(OD\right)-\log\left(OD_{0}\right)\right)}{\left(t-t_{0}\right)}$$

To test for significant differences in biomass and specific growth rate among the treatments, ANOVA was performed in R Studio. A post hoc Tukey HSD was used to isolate significant treatment means while applying a correction to account for multiple comparisons.

### 4.3. Analysis of Volatile Metabolites

Fermentations that had reached stationary phase were analysed for volatile compounds released into the headspace. Yeast cells were pelleted by centrifugation (3000× *g*, 5 min), and the supernatant was stored at −20 °C prior to volatile metabolite analysis.

#### 4.3.1. Sample Preparation for Volatile Metabolite Analysis

Prior to the analysis, samples were defrosted at 4 °C, aliquoted (5 mL) into a 20 mL amber glass vial, and internal standard 12-bromo-1-dodecanol (5 µL, 10 mM) was added to each. Vials were fitted with an 18-shore PTFE septum and cap.

#### 4.3.2. Headspace Solid-Phase Microextraction (HS-SPME)

Extraction of volatile compounds from the sample headspace was achieved using a SPME fibre coated with DVB/CAR/PDMS and with a film thickness of 50/30 μm (Supelco 57329-U). Samples were incubated (60 °C, 10 min) with agitation at 200 rpm. Following incubation, the SPME fibre was exposed to the vial headspace (10 min) to allow absorption of the compounds.

#### 4.3.3. GC-MS Analysis

Following HS-SPME extraction, the fibre was injected into an Agilent 7890B gas chromatograph coupled with a 5977A inert mass spectrometer (GC–MS) under splitless mode at 250 °C and held in the injection port for 1 min to allow complete desorption of analytes. The fibre was held at 250 °C for a further 1 min to ensure the fibre was clean for subsequent analyses. The GC oven started at an initial temperature of 35 °C. Following injection, the temperature was raised at 4.5 °C·min^−1^ to 180 °C. The temperature was then raised at 40 °C·min^−1^ to 160 °C. Finally, the temperature was raised to 260 °C at 10 °C·min^−1^. The carrier gas used was helium (He) at 7.7 psi with a flow rate of 1.09 mL·min^−1^ through a Rtx-5MS column with a coating thickness of 0.25 μm, diameter of 0.25 mm, and total length of 30 m. Interface temperature was held at 250 °C and quadrupole temperature was at 200 °C. The ion source was operated in electron impact ionisation mode at 70 eV. Compounds were detected using mass spectra acquired in scan mode in the range of 33 to 400 m/z.

#### 4.3.4. Data Analysis

Raw GC–MS data generated from SPME analysis were processed through the Automated Mass spectral Deconvolution and Identification System (AMDIS) software. Identification of compounds was based on the National Institute of Standards and Technology (NIST) 2017 mass spectral library, only considering those with a match quality above 90%. Compounds were considered tentatively identified (putative identification). The R package MassOmics was used for automated integration of reference ion peak area [41]. Each identification was individually screened, and manual retention time correction and subsequent reintegration were performed where required. To enable comparisons between samples, the data were normalised by the internal standard peak intensity and log-transformed to fit normal distribution criteria for downstream statistical analyses. Two silence samples were removed due to mishandling during the sample preparation stage. The relative abundances of each compound were analysed for differences between sound treatments using ANOVA followed by post hoc Tukey multiple comparisons to isolate significant means whilst using a correction for *p*-values for multiple comparisons. To attempt to distinguish samples among classes (sound treatment), a linear discriminant analysis (LDA) was used as LDA maximises separation between classes, rather than a PCA which maximises variation in the reduced dimension [47]. We applied LDA to visualise samples using the top 20 identified common aroma compounds, by relative abundance, detected across all samples and all conditions. The LDA results were then validated using leave-one-out cross-validation (LOOCV) [13]. All statistical analyses were carried out using RStudio 3.4.1.

## 5. Conclusions

This study demonstrates that audible sound stimuli can significantly alter *S. cerevisiae* growth and the volatile compounds it produces. The growth rate of yeast in fermentations exposed to sound was increased, with a 28.9% higher specific growth rate in fermentations exposed to high-frequency (10 kHz) sound compared to silence. Furthermore, the volatile profile of the resulting ferments was modulated by the sound frequency applied. Of particular interest was the alteration of levels of aroma compounds produced in the synthetic liquid medium by the different frequencies of sound. The levels of citrus compounds, such as limonene (orange) and isocyclocitral (leafy citrus), were increased by both sound treatments, whereas sweet fruit compounds, such as ethyl hexanoate (banana) and ethyl octanoate (apricot), decreased. The production of benzoin (vanilla) was elevated by the application of low-frequency sound, yet decreased by high-frequency treatment. Overall, these results suggest that specific audible sound stimuli could be employed by beverage fermentation industries to increase productivity and achieve desired aroma profiles, thus providing a novel avenue with which to diversify product styles.

## Figures and Tables

**Figure 1 metabolites-11-00605-f001:**
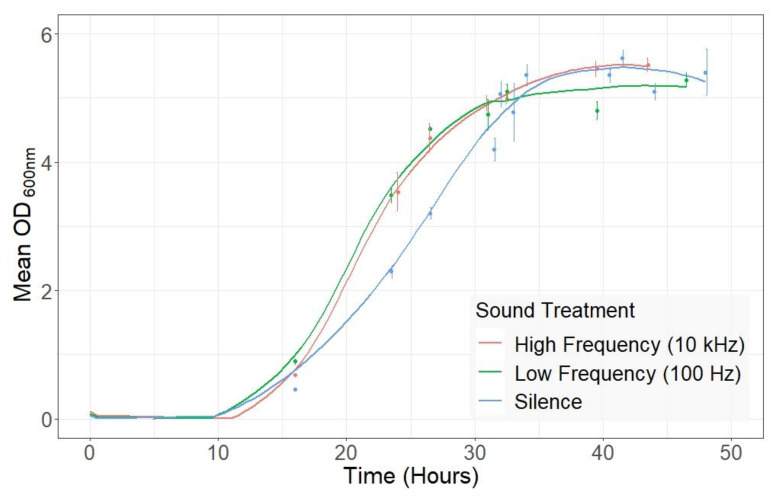
Growth curves of *Saccharomyces cerevisiae* exposed to different sound conditions. Two growth experiments under control silence conditions were averaged, shown in blue. Growth curves exposed to high-frequency sound of 10 kHz (red) and low-frequency sound of 100 Hz (green) are presented. Plotted values are means of biological replicates (n = 3) from each time point and error bars (±Standard Error).

**Figure 2 metabolites-11-00605-f002:**
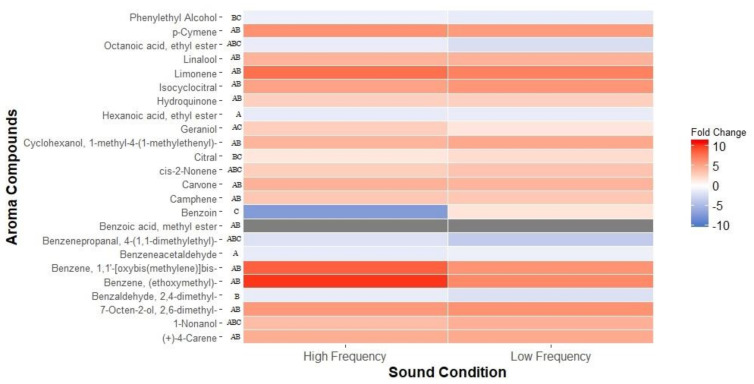
Fold changes of aroma-associated volatile metabolite levels produced when exposed to high-frequency (10 kHz) and low-frequency (Hz) sound when both are compared to silence. Data are displayed as a heat-map (right), with red-shade indicating increased abundance blue-shade indicating decreased abundance (compared to silence). The pairwise significance for each compound was assessed using a post-hoc Tukey ANOVA (*p* < 0.05); A- high-frequency versus silence, B- low-frequency versus silence, C- high-frequency versus low-frequency. Methyl benzoate, grey, was outside the range of our scale, with significantly higher abundance for both high-frequency and low-frequency sound treatments (33- and 30-fold, respectively).

**Figure 3 metabolites-11-00605-f003:**
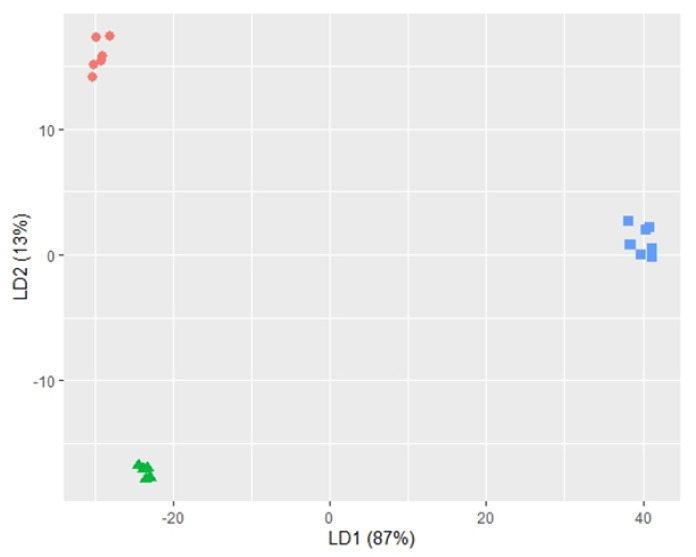
Aroma compounds identified from SPME GC-MS successfully separated the different sound treatments. Log-transformed relative abundance data of the 24 putatively identified aroma compounds from each ferment are projected onto two-dimensional space (following normalisation against an internal standard), resulting in three distinct clusters. Red = high-frequency (10 kHz), green = low-frequency (100 Hz), blue = silence.

**Figure 4 metabolites-11-00605-f004:**
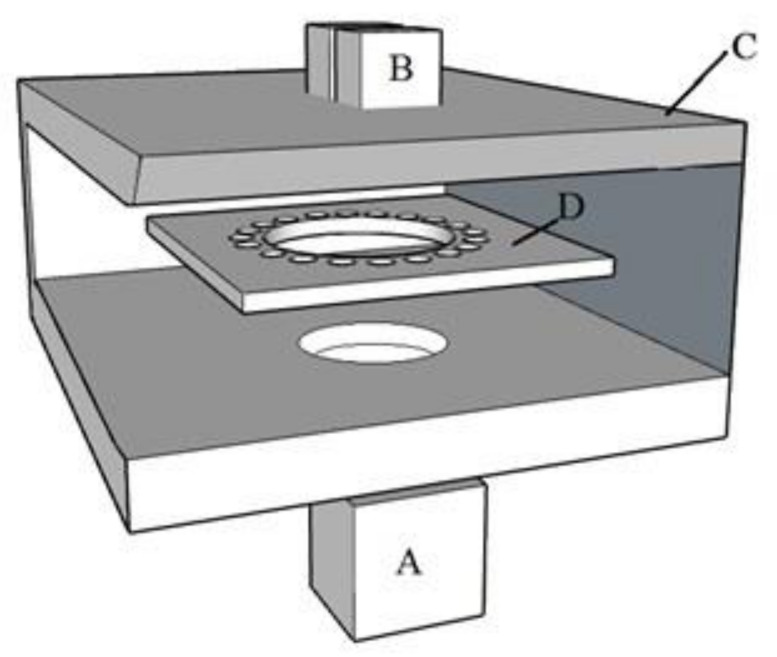
Sound Isolation Chamber used to house the fermentations and apply sound stimuli. The entire setup was placed inside a temperature control room at 28 °C and allowed to acclimatise. (**A**) Low-frequency subwoofer. (**B**) Two middle-high frequency speakers. (**C**) Sound isolation chamber, cross-sectioned to show internal position of fermentations and the experimental 22 mm thick medium density fibreboards that act as soundproofing walls, lowering the internal sound intensity. (**D**) Position of the circular fermentation tube rack, which ensures equal sound intensity within each fermentation.

**Table 1 metabolites-11-00605-t001:** Classification table obtained from leave-one-out cross-validation of 20 samples containing 24 aroma-associated volatile metabolites.

Original Source	Predicted Source		
	High-Frequency	Low-Frequency	Silence
High-frequency	6	0	0
Low-frequency	0	6	0
Silence	0	0	8

High-frequency = 10 kHz; Low-frequency = 100 Hz; Silence = no sound stimulus.

## Data Availability

The data presented in this study are available in article and Appendix A.

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
