# Peer review of "Sound Stimulation Can Affect Saccharomyces cerevisiae Growth and Production of Volatile Metabolites in Liquid Medium"

_metabolites, 2021, doi:10.3390/metabo11090605_

Round 1

Reviewer 1 Report

The article provides interesting data on the effect of sound on the growth rate of yeast, biomass yield and the level of production of volatile compounds. The composition of volatiles produced during fermentation was shown to be markedly altered by the application of sound. The results obtained may be important for the food industry, since the composition of volatile compounds gives most of the taste to fermented beverages and food products.

Some questions, the answers to which should be included in the discussion:

  1. As follows from the text ( Lines 66-68): Consequently, applying sound to alter the growth rate, thereby altering the length of the exponential growth phase, may result in changes to the volatile metabolite composition produced during fermentation.

In this regard, the question arises: Can a change in the growth rate and the length of the exponential growth phase, due to the composition of the medium, lead to similar changes in the composition of volatile metabolites formed during fermentation?

  1. Is there an explanation for the significant increase in the growth rate of Brevibacterium sp. compared to yeast?
  2. The authors write that the application of sound can directly affect the cell membrane, leading to physiological stress, which can, in turn, alter the production of secondary metabolites.

Are there any data on changes in the yeast transcriptome when they are exposed to sound?

Author Response

The article provides interesting data on the effect of sound on the growth rate of yeast, biomass yield and the level of production of volatile compounds. The composition of volatiles produced during fermentation was shown to be markedly altered by the application of sound. The results obtained may be important for the food industry, since the composition of volatile compounds gives most of the taste to fermented beverages and food products.

  1. Some questions, the answers to which should be included in the discussion: As follows from the text ( Lines 66-68): Consequently, applying sound to alter the growth rate, thereby altering the length of the exponential growth phase, may result in changes to the volatile metabolite composition produced during fermentation. In this regard, the question arises: Can a change in the growth rate and the length of the exponential growth phase, due to the composition of the medium, lead to similar changes in the composition of volatile metabolites formed during fermentation?

Changes in the medium are likely also to produce changes in the volatile profile. This is why we used a malt extract media so that the fermentations were similar to beer and industrial applications. We also used a single batch of media to ensure there were no differences between our ferments deriving from the media, this has been emphasized further in our revised version of the manuscript. (Line 79)

  1. Is there an explanation for the significant increase in the growth rate of Brevibacterium sp. compared to yeast?

The underlying mechanism driving the observed changes in microbial growth when exposed to sound is unclear, and as such, it is difficult to determine why the response could vary between microbial species. This point has been emphasized in the revised manuscript (Line 181).

  1. The authors write that the application of sound can directly affect the cell membrane, leading to physiological stress, which can, in turn, alter the production of secondary metabolites. Are there any data on changes in the yeast transcriptome when they are exposed to sound?

There are no published studies investigating the effect of sound on the yeast transcriptome. However, one published study has shown that sound increased RNA production in E. coli, a reference to which has been included in revisions of the manuscript at Line 206.

Reviewer 2 Report

- Please underline the importance of S. cerevisiae as a model organism in microbiology and in eukaryote biology. This gives further relevance to your study, particularly in consideration of the studies already done in E. coli. 

https://www.mdpi.com/2218-1989/6/1/8

https://www.mdpi.com/2073-4409/7/2/14 

- Please consider the following article to integrate your literature about the topic. This can be of help to improve introduction/discussion, particularly underlining the relevance of this kind of studies, in terms of potential applications. 

https://doi.org/10.3389/fevo.2021.662588

https://www.mdpi.com/2076-0817/10/1/63

https://doi.org/10.1016/j.apacoust.2020.107620

https://doi.org/10.1111/jfs.12856

https://doi.org/10.1002/jctb.5857

https://peerj.com/articles/1920/ 

- Line 9: on microorganisms 

- low frequency, high frequency, low intensity...please consider to insert a hypen

- Lines 56-65: The metabolism of microorganisms is important for many industrial processes, including the production of a wide variety of food products and high-value molecules [16]. Of these processes, the production of bread and alcoholic beverages through fermentation by S. cerevisiae remains one of the largest industrial uses of this microorganism (https://www.mdpi.com/2306-5710/2/4/30, https://www.mdpi.com/1420-3049/21/4/483). The production of secondary metabolites by yeast, such as higher alcohols, esters, and phenols, impacts the flavour profile of fermented products [17]. The compounds produced are determined by the strain of yeast used, the selected raw matrices, interaction among yeasts, growth rate, and environmental conditions during fermentation [18, https://www.mdpi.com/2311-5637/6/2/55]. 

- Line 70: fermentation. Both strain and media selections were inspired to beer production.

- Line 131: produced by S. cerevisiae

- Line 189: may, in turn, 

- Line 203: hops, cereals, yeast/bacteria and adjuncts (https://www.mdpi.com/2304-8158/10/8/1831). 

Line 204: to mimic brewing

- Among the future perspectives, please include the interest in conducting this kind of study also for non-Saccharomyces (https://www.mdpi.com/1420-3049/26/3/644). 

- Among the future perspectives, can be interesting to include the potential application of analytical approaches that allow on line monitoring of VOCs associated with the cultures (ref needed)

- Table A1: Associated Odour Description

Author Response

Please underline the importance of S. cerevisiae as a model organism in microbiology and in eukaryote biology. This gives further relevance to your study, particularly in consideration of the studies already done in E. coli.

There are a number of publications, including reviews, that highlight the benefits of working with model organisms, such as S. cerevisiae, due to the depth of existing knowledge of their behaviour and metabolic functioning. In addition, S. cerevisiae is particularly important in an industrial setting for fermentation productions.  We have revised the manuscript to cite key publications to support our assertion of the value of using this yeast species as a model organism, i.e., https://www.mdpi.com/2218-1989/6/1/8 & https://www.mdpi.com/2073-4409/7/2/14 (see lines 56 - 64).

- Please consider the following article to integrate your literature about the topic. This can be of help to improve introduction/discussion, particularly underlining the relevance of this kind of studies, in terms of potential applications.

Thank you for this helpful suggestion, we have proceeded to include all of the recommended publications in our revisions to our manuscript, i.e., https://doi.org/10.3389/fevo.2021.662588 provides a helpful review in the topic area,https://www.mdpi.com/2076-0817/10/1/63 helps to emphasise the non-Saccharomyces input, and https://doi.org/10.1016/j.apacoust.2020.107620 helps with demonstrating that sound increases E. coli motility. We have also included https://doi.org/10.1111/jfs.12856, https://doi.org/10.1002/jctb.5857, https://peerj.com/articles/1920/.
Most of these publications have also been included in our revised discussion. In particular, they were helpful to emphasize non-Saccharomyces in wine production and highlight some potential mechanisms in E. coli that could translate to yeast but have not been tested yet.

- Line 9: on microorganisms 

We have revised accordingly.

- low frequency, high frequency, low intensity...please consider to insert a hypen

This has been included in the revised manuscript as suggested.

  • Lines 56-65: The metabolism of microorganisms is important for many industrial processes, including the production of a wide variety of food products and high-value molecules [16]. Of these processes, the production of bread and alcoholic beverages through fermentation by S. cerevisiae remains one of the largest industrial uses of this microorganism (https://www.mdpi.com/2306-5710/2/4/30, https://www.mdpi.com/1420-3049/21/4/483). The production of secondary metabolites by yeast, such as higher alcohols, esters, and phenols, impacts the flavour profile of fermented products [17]. The compounds produced are determined by the strain of yeast used, the selected raw matrices, interaction among yeasts, growth rate, and environmental conditions during fermentation [18, https://www.mdpi.com/2311-5637/6/2/55]. 
    We have revised the manuscript to better emphasize the narrative advanced by the reviewer. These additional references suggested by the reviewer have been included in our revised manuscript.

- Line 70: fermentation. Both strain and media selections were inspired to beer production.
This connection has been included in the revisions to the manuscript.

- Line 131: produced by S. cerevisiae
This has been included in the revisions to the manuscript.

- Line 189: may, in turn, 
This has been included in the revisions to the manuscript.

- Line 203: hops, cereals, yeast/bacteria and adjuncts (https://www.mdpi.com/2304-8158/10/8/1831).
This suggested reference and edits to the text have been included in the revised manuscript.

Line 204: to mimic brewing
This has been included in the revised manuscript

- Among the future perspectives, please include the interest in conducting this kind of study also for non-Saccharomyces (https://www.mdpi.com/1420-3049/26/3/644). 
This point has been included (lines 230 -238) in the revisions to the manuscript, in particular emphasising the non-Saccharomyces in wine production which affects flavour.

- Among the future perspectives, can be interesting to include the potential application of analytical approaches that allow on line monitoring of VOCs associated with the cultures (ref needed)
We have revised the manuscript to refer to the use of online monitoring methods for volatile organic compounds in our revisions of the manuscript, i.e., doi:10.1128/AEM.02069-07.

Round 2

Reviewer 2 Report

The authors adequately addressed the criticisms. A few points to be considered in the next revision stage. 

Lines 233-34: It would be interesting to assess if sound could...

Line 234: metabolosim?

Line 239: alertering?